# A free energy landscape analysis of resistance fluctuations in a memristive device

Sebastian Walfort [1] ✉, Xuan Thang Vu [2], Jakob Ballmaier[1], Nils Holle [1], Niklas Vollmar [1] & Martin Salinga [1] ✉

Resistance noise in memristive devices is often attributed to simple thermally activated processes, such as fluctuations across single energy barriers. However, this picture may underestimate the complexity of the underlying atomic dynamics, which can be described as transitions between many local minima in a high-dimensional free energy landscape shaped by energetic and entropic contributions, yet such landscapes are difficult to access experimentally. Using a hidden Markov model, we analyse resistance fluctuations in a nanoscopic volume of the phase-change material germanium telluride. We quantify the transition rates between discrete resistance states over a wide temperature range. The rates follow an Arrhenius-like behaviour, but the extracted attempt frequencies span several orders of magnitude and include values far below typical phonon frequencies. This spread reflects substantial entropic contributions to the free energy barriers, which we quantify by tracking individual transitions across temperatures. This approach should be broadly applicable to memristive materials, where significant resistance changes are linked to atomic-scale transitions.

Memristive devices can retain a memory of past voltage pulses because the accompanying currents, fields or heat alter the magnetic or atomic order, resulting in a measurable change in electrical resistance. This tunable, non-volatile contrast between configurations is desirable for data storage and for non-von Neumann computing approaches[1]. Technologically mature examples include magnetic[2], resistive[3] and phase-change[4,5] random-access memories.

Common to these technologies is that continued development has pushed devices to dimensions in which only a countable number of atoms define a resistance state. At such scales, the resistance becomes highly sensitive to small structural rearrangements, and substantial read noise can arise from thermal fluctuations between configurations. Desired and undesired transitions are commonly modelled as thermally activated processes limited by an activation energy. The underlying energy landscape is often reduced to a one-dimensional double-well potential with two minima separated by a barrier. Although this can be accurate for, for example, an isolated defect in a large crystalline volume, it may not capture the structural complexity of real devices far

from thermodynamic equilibrium. In particular, it neglects entropic contributions that can shape the free energy landscape and influence state occupancy and transition rates.

Among memristive material systems, phase-change materials (PCMs) offer a high contrast between highly resistive amorphous states and relatively conductive crystalline counterparts. Combined with their highly nonlinear crystallization kinetics that enable rapid tuning of amorphous-to-crystalline volume fractions, PCMs become useful in dense crossbar arrays for neuromorphic and in-memory computing[6] and for embedded memory[7]. The amorphous state is created by a short electrical pulse that melts a small volume, which then quenches before crystallization can occur. This process is a specific instance of a more general phenomenon: during cooling, the structural relaxation time of the supercooled liquid increases drastically with decreasing temperature. When the atomic structure no longer has time to assume the equilibrium supercooled liquid structure at a given cooling rate, it falls out of equilibrium, undergoing a glass transition to an amorphous solid state. Understanding this transition and the associated glassy

[1]University of Münster, Institute of Materials Physics, Münster, Germany. [2]RWTH Aachen University, Institute of Materials in Electrical Engineering 1, Aachen, Germany. ✉e-mail: sebastian.walfort@uni-muenster.de; martin.salinga@uni-muenster.de

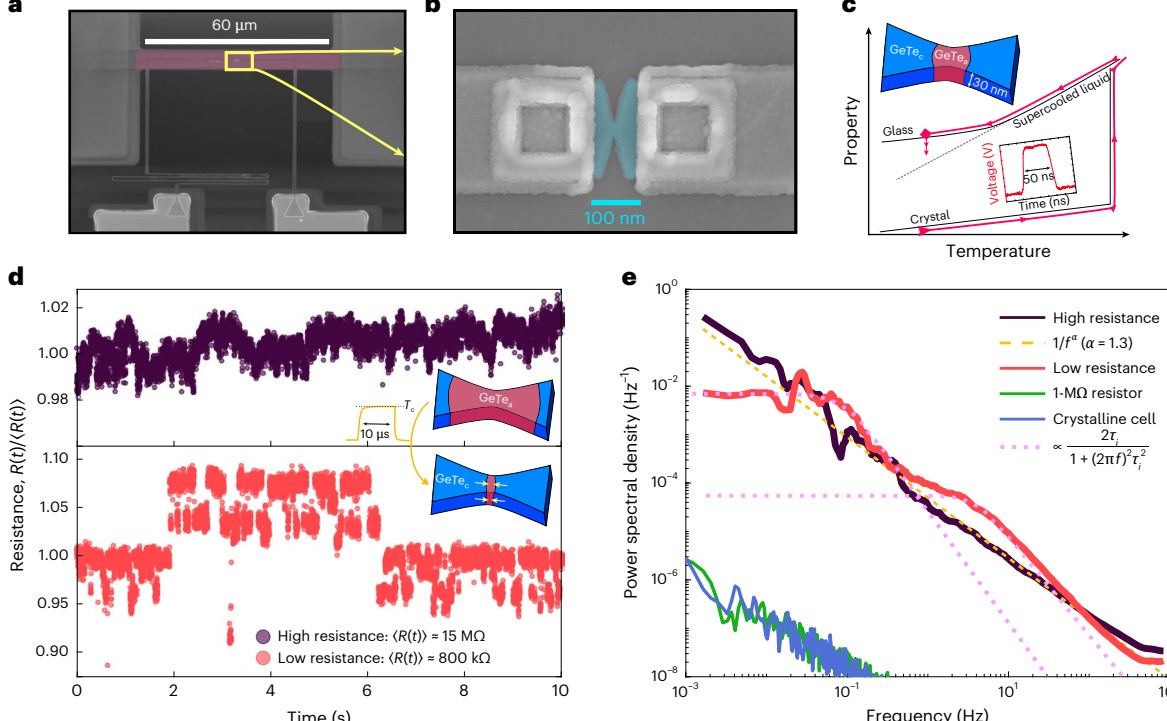

**Fig. 1 | Experimental design. a**, Scanning electron microscopy image of the device architecture. A 60-μm-long platinum microheater (pink) is fabricated on top of 200-nm silicon dioxide and subsequently buried underneath an additional 150 nm of dielectric oxide. **b**, The GeTe cell is placed in the centre of the microheater. It consists of a GeTe volume (blue) contacted by tungsten electrodes and encapsulated in a dielectric oxide. **c**, Sketch of the glass formation process. A voltage pulse heats the crystalline centre (GeTe$_c$) volume above the melting temperature, which then quenches to the glass state (GeTe$_a$)

and relaxes towards the supercooled liquid. **d**, Normalized time-domain resistance fluctuations of a large glass volume (15-MΩ cell; dark purple) and a small glass volume (800-kΩ cell; bright red). The small glass volume is formed by partial recrystallization through the application of short heating pulses. **e**, Corresponding frequency-domain power spectra together with the spectrum of a fully crystalline cell (100 Ω) and a 1-MΩ carbon film resistor for comparison. As a guide to the eye, a $1/f^{1.3}$-type spectrum (dashed yellow) as well as two Lorentzians with $\tau_i \approx \{0.15\ \text{s}, 10\ \text{s}\}$ (dashed pink) are also plotted.

dynamics remains a major challenge in condensed-matter physics, with relevance extending well beyond PCMs[8,9].

A central challenge in using PCMs as electronic memory arises exactly because the amorphous states are inherently out of equilibrium. As the glass evolves towards the supercooled liquid, the resistivity increases over time (resistance drift)[10], which must be accounted for in read-out and inference[11]. Furthermore, it is believed that the same structural fluctuations that drive resistance drift can also give rise to excessive low-frequency noise, limiting learning accuracy in neuromorphic systems[12]. Mitigation typically relies on hardware-aware retraining, which adds overhead and only partially alleviates these issues[13].

A general theoretical framework from glass theory describes dynamics as motion on a high-dimensional energy landscape with many local minima[14–16]. When barriers between minima exceed $k_\text{B}T$, Goldstein's potential energy landscape picture predicts two timescales: fast fluctuations within a basin (inherent structure[17]) and rare thermally activated transitions between basins[18]. This view is widely supported by simulations of model glass formers[19,20], but is difficult to verify experimentally, especially in disordered systems in which spatial and temporal averaging obscures individual transitions[21].

This inability to resolve distinct features of the energy landscape in experiments has limited the understanding of resistance drift and noise in PCMs. Drift typically appears as a continuous increase in resistivity over many orders of magnitude in time, following a power law. Similarly, electrical noise tends to show a featureless $1/f$ spectrum. These observations have led to contrasting interpretations: drift has been attributed to both collective structural relaxation across the

entire amorphous volume and to localized processes involving isolated defects[22,23]. Likewise, $1/f$ noise can be modelled as a broad distribution of double-well potentials, consistent with an unresolved superposition of many independent fluctuators[24,25].

Here we show that individual transitions between local minima or (meta)basins can be resolved in time by monitoring resistance fluctuations in a nanoscopic PCM volume. Similar approaches have been used to study noise dynamics in small systems with varying levels of disorder[26,27]. Germanium telluride (GeTe) serves as a model system because its electrical properties are highly sensitive to structural rearrangements. Although fast quenching is required to form the glass, small glass volumes remain stable for years, enabling the repeated probing of the same landscape region over a range of temperatures. This allows us to quantify energetic and entropic contributions to landscape barriers and basins, revealing a complex free energy landscape governing the structural/noise dynamics.

## Observing states in a fluctuating resistance

The high quenching rates are realized by confining GeTe to a nanoscopic volume and applying short (self-)heating pulses. The device integrates a buried microheater beneath the GeTe cell (Fig. 1a,b). A 50-ns pulse melts a central fraction of initially crystalline GeTe, which then quenches to the glass state and increases the cell resistance from 100 Ω to about 10 MΩ. The resulting high-resistivity glass volume consequently governs the electrical response.

Two observations are typically made after glass formation in PCMs. Both can be interpreted as signatures of an underlying energy landscape. First, the time-averaged resistivity increases during physical ageing towards the equilibrium supercooled liquid[22,23]. Second, the

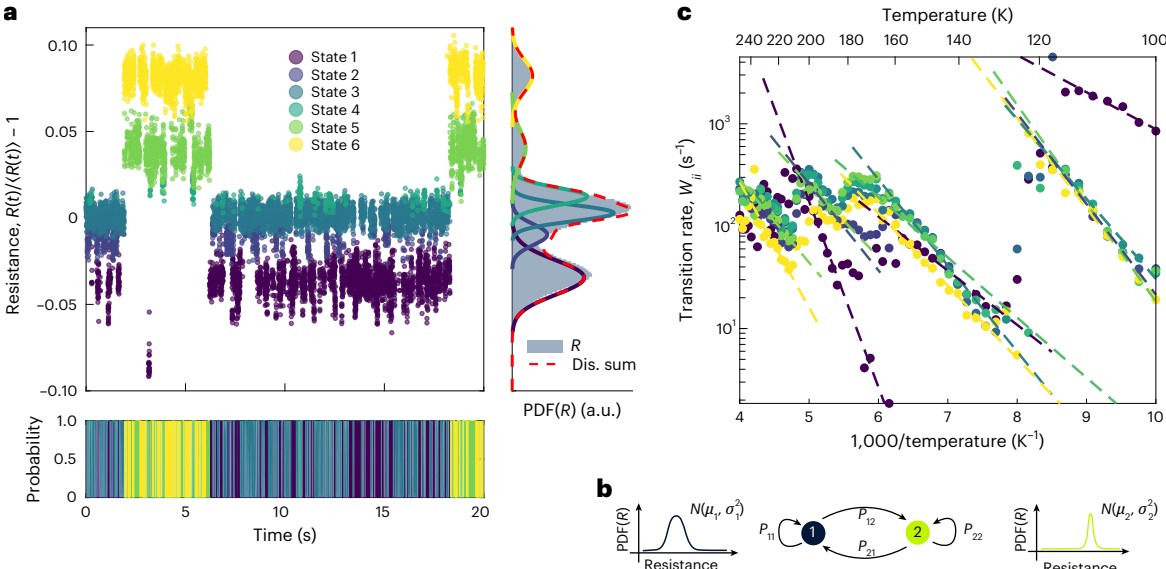

**Fig. 2 | Modelling of noise dynamics. a**, Markov model fit with six states. Data points are assigned to states based on the most likely state sequence through the time series. The probability that a data point belongs to a state is colour coded below. The distribution of resistance values ($R$) over the 1,000-s-long measurement is shown (in grey) on the right (PDF, probability density function), together with the normal distributions of the states and their sum (Dis. sum).

The states are sorted from low (blue) to high (yellow) relative resistances. This colour code is used throughout the manuscript. **b**, Depiction of a two-state hidden Markov model with normally distributed resistances as emissions from the states and transition probability matrix $P$. **c**, State transition rate (inverse state lifetime) is plotted against inverse temperatures. The dashed lines are Arrhenius equation fits.

resistivity fluctuates around this drifting mean with a noise power spectral density orders of magnitude larger than in the crystalline phase[24,25]. Here we focus on the latter.

Figure 1d shows the normalized time-domain resistance fluctuations and Fig. 1e shows the corresponding power spectra. For a high-resistance cell (large glass volume), the spectrum follows an approximate 1/$f$ dependence and exceeds the crystalline noise by more than five orders of magnitude. Such spectra can arise from a superposition of Lorentzians $S_i(f)$ from a large number of fluctuators, with time constants $\tau_i$ and a uniform distribution of energy barriers $E_i$ (refs. 28,29):

$$S_i(f) \propto \frac{2\tau_i}{1+(2\pi f)^2 \tau_i^2}, \quad \tau_i \propto \exp\left(\frac{E_i}{k_B T}\right). \quad (1)$$

These fluctuators may originate from low-energy excitations such as two-level systems[21,30] or double-well potentials[25]. In the landscape picture, the high-resistance time series reflects unresolved fluctuations during transitions between resistance states (basins), consistent with many contributing fluctuators.

By applying consecutive heating pulses through the buried microheater, a fraction of the glass volume is recrystallized each time in a controlled manner. The noise characteristics change when the resistance of the GeTe cell is reduced by a factor of 10–50. The power spectrum is now well described by only two Lorentzians and the time trace fluctuates between a small number of states, with a clear separation of slow and fast timescales. Apparently, the external heating pulses primarily reduce the active glass volume rather than annihilating defect states, similar to small glass volumes formed by low-amplitude amorphization pulses that exhibit comparable noise characteristics (see discussion around Fig. 5), as also observed elsewhere for a different PCM[31].

The largest observed change in relative resistance between discrete states is about 10% (that is, 100 kΩ for a 1-MΩ cell), which is not surprising for PCMs. In the high-temperature metallic liquid, a Peierls distortion structural motif of the crystalline phase is (partially) suppressed, but it increases continuously during supercooling, accompanied by the opening of a pseudogap and slowing dynamics[32]. This

evolution is interrupted by the glass transition. During physical ageing, the structure continues to evolve, the density of states at the Fermi level decreases, and the resistivity increases (resistance drift)[22,33]. In our GeTe cells, the resistance rises from 30 kΩ to 3 MΩ within the first second after melt quenching and continues to increase for weeks, following a power law[10]. The same structural changes probably underlie the large resistance fluctuations, as further explored in Fig. 5.

The indirect heating pulses not only induce partial recrystallization but also momentarily accelerate physical ageing of the glass. The system seems to become trapped in a low-energy region of the landscape and does not escape even under subsequent moderate heating. Accordingly, no hysteresis is observed in the average resistance or in the noise dynamics during the subsequent temperature-dependent studies. In summary, for small glass volumes, the resistance time series shows infrequent jumps between discrete states. Each state is characterized by a mean resistance and rapid fluctuations around it. In the landscape picture, this is consistent with fast exploration within a (meta)basin (with barriers negligible compared with $k_B T$) and rare transitions to neighbouring basins across larger barriers.

## Markov modelling of noise dynamics

For the following analysis of the landscape characteristics, we first need a method to identify states in the resistance time series based on a probabilistic argument. The separation of timescales in the landscape picture as well as in the experimental data suggests a hidden Markov model as an appropriate statistical description for the noise dynamics. In this model, the system transitions through a sequence of states in a memoryless (Markov) process, that is, with transition probabilities for the next state in a sequence that depend only on the current state[34,35]. The hidden state sequence cannot be observed directly. Instead, the emissions (here the observed resistance values) from the states are modelled with some state-specific probability density function (the resistance emissions are assumed to be normally distributed; Fig. 2a, right). The fact that the hidden Markov model captures both emitted resistance distributions and, more importantly, the time series character of the measurement, is crucial for identifying states in the 'noisy' data. Figure 2b illustrates

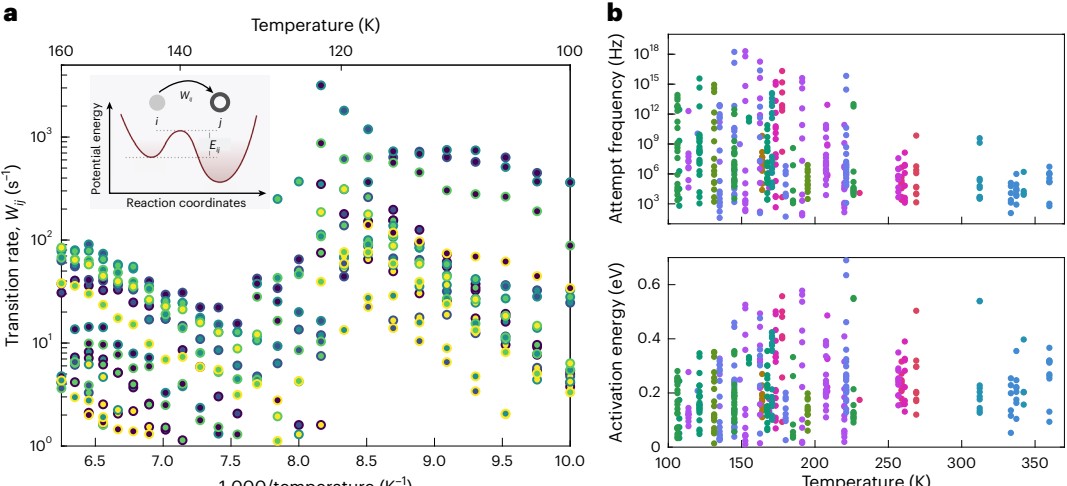

**Fig. 3 | Beyond state lifetimes. a**, Transition rates between states against inverse temperatures. Arrhenius fit lines are omitted for clarity. Each point corresponds to a directed transition $i \rightarrow j$ observed at a certain temperature. The system transitions from state $i$ assigned to the face colour of the circle to state $j$ assigned to the edge colour. The colour code for sorting from low to high resistances is the same as that shown in Fig. 2. The inset sketches a landscape with an asymmetric activation energy. A visualization of an asymmetry in the attempt frequency is less obvious. **b**, Pairs of attempt frequencies and activation energies from Arrhenius equation fits plotted against the average of the temperature interval in which the corresponding states were observed. The different colours correspond to different measurement series. Between two series, the GeTe cell is crystallized completely and re-amorphized, varying the size of the amorphous volume. The spread of points illustrates the breadth of activation energies, particularly the wide distribution of attempt frequencies observed across all series. For uncertainty estimates, Supplementary Fig. 2 shows the corresponding activation energy–attempt frequency pairs together with their 95% confidence ellipses. The size of the amorphous volume is not correlated with the observed activation energies and attempt frequencies (Supplementary Fig. 3).

the model parameters for a two-state system consisting of the two normal distributions for the state emissions and the 2 × 2 transition probability matrix.

The only user input to the model is the number of states. With a time resolution of ~1 ms and 1,000-s-long observation windows, the noise dynamics are well described using four- to six-state models. Figure 2a shows an excerpt of the results of fitting a model with six states to the resistance time series of the small glass volume from above. The data points are assigned to states based on the most probable state sequence through the time series. A visual inspection of the fit results suggests that the model can describe the noise dynamics, in the sense that the distribution functions fit the emitted resistance values and the transition probability matrix captures the observed state sequence. As with more standard curve-fitting procedures, this qualitative statement can be checked by calculating the (pseudo)residuals[34]. Although the decision on the exact number of states is rather inconsequential for further analysis, the pseudoresiduals (Supplementary Fig. 1) can also give an indication for the minimum number of states.

We measure the noise dynamics of a small glass volume over a wide temperature range and fit hidden Markov models to each time trace. At each temperature, states are sorted from low to high relative resistance. The transition rate (inverse average lifetime) is plotted versus inverse temperature (Fig. 2c). The same states, with characteristic lifetimes and relative resistances, persist across several temperatures, and their transition rates increase exponentially with temperature, consistent with thermally activated kinetics with approximately temperature-independent barriers. This is expected even though GeTe is fragile in the supercooled liquid[36]: individual landscape barriers are temperature independent, and below the glass transition, the dynamics of fragile glass formers are commonly described by an Arrhenius activation energy[37]. For instance, crystallization kinetics in PCMs below the glass transition follow an Arrhenius law with a single activation energy[38].

When the transition rate exceeds the ~1-kHz upper limit of the experimental time resolution, states leave the observation window and new states enter, governing the noise dynamics through the next temperature interval. Fitting an Arrhenius equation

$$\nu = \nu_0 \exp\left(-\frac{E_a}{k_B T}\right), \tag{2}$$

with attempt frequency $\nu_0$, activation energy $E_a$, Boltzmann constant $k_B$ and temperature $T$, to the inverse state lifetimes, only yields an effective energy barrier, corresponding to a weighted average of the transition barriers to all other accessible states. We can avoid this coarse description of the landscape, because the Markov model fit provides us with access to the transition rate between any two states. To this end, we calculate the transition rate matrix from the transition probability matrix and the average state lifetimes. Figure 3a plots the off-main-diagonal elements of the transition rate matrix against inverse temperatures. The transition rates can again be followed across several temperatures and show thermally activated behaviour. Overall, the rates tend to be highly asymmetric. There can be an order of magnitude difference between the transition rate from state $i$ to state $j$ compared with the opposite direction. Arrhenius equation fits, therefore, give a set of two different attempt frequencies and activation energies for the same barrier (Fig. 3a, inset).

To probe different sections of the landscape, we repeat the measurement and analysis several times. Between series, the GeTe cell is fully crystallized and re-amorphized. Figure 3b plots the resulting ($\nu_0$, $E_a$) pairs against the average temperature of the interval in which the corresponding transition was observed. Because barriers are not sampled uniformly across temperature, no definitive temperature trend can be inferred, although higher temperatures tend to sample slightly larger $E_a$ and lower $\nu_0$. Correlations between $E_a$ and $\nu_0$ can arise from the finite time resolution of the experiment (Supplementary Figs. 2 and 3).

The activation energies span 0.02 eV to 0.6 eV, in good agreement with potential energy barriers encountered during the physical ageing of GeTe glass in atomistic (nudged elastic band) simulations[22]. Simulations of amorphous GeTe and closely related materials[39–41] also report electronic gap states with depths in a similar energy range. Here we remain agnostic about the microscopic identity of the states and barriers, as we are convinced that electronic and atomic degrees of freedom are strongly intertwined. The total energy of the glass system includes electronic kinetic and exchange–correlation contributions as well as

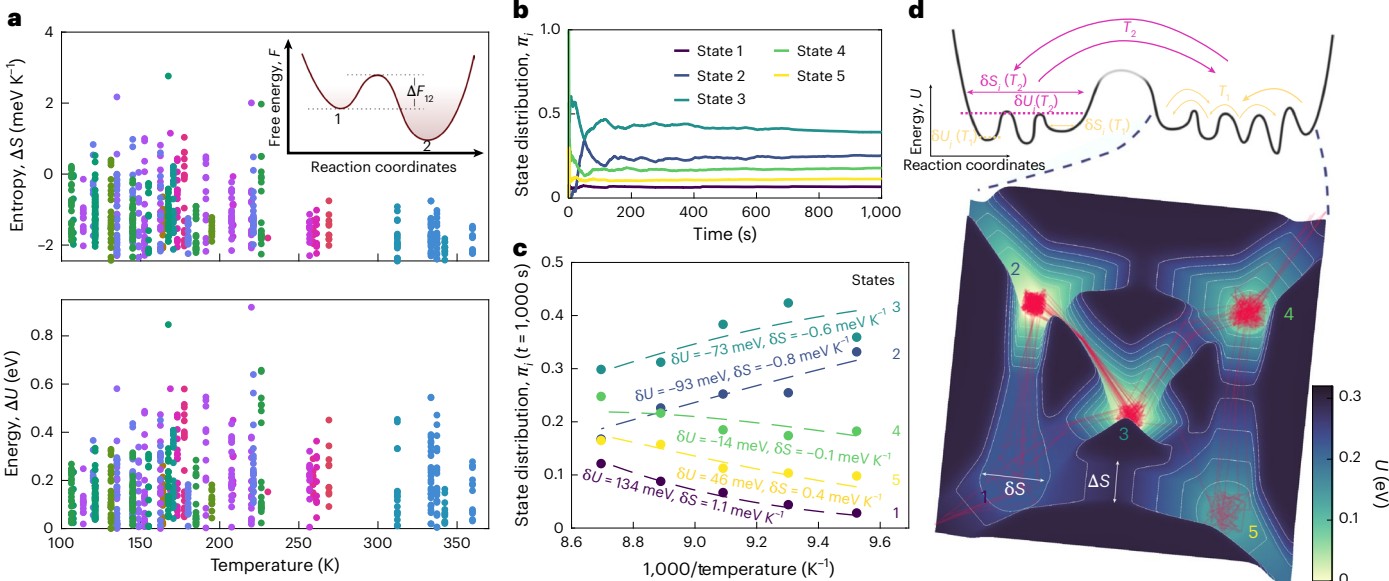

**Fig. 4 | Plotting the landscape. a**, Pairs of entropy and energy contributions to free energy barriers, as sketched in the inset. **b**, Example of the time dependence of the state distribution function. The state distribution function converges within the experimental observation window. **c**, Example of the temperature dependence of the state distribution function together with a Boltzmann distribution fit using the (relative) state entropies and energies. **d**, Bottom: visualization of a landscape based on the derived energies (height of the landscape) and entropies (mapped to widths) of barriers and basins. The red lines mark the experimentally observed path of the system through the landscape over a 2-s time window (Supplementary Video 1). Top: the system samples different (free) energy barriers at higher temperatures ($T_2 > T_1$), with, on average, a small increase in internal energy and entropy difference between the valley and saddle point.

Coulomb interactions among electrons and ions, and none of these terms can generally be assumed negligible or independently fluctuating. Consistent with this view, the nudged elastic band simulations of ageing in GeTe show barriers comparable with the energies of electronic states formed or annihilated near the Fermi level[22]. The wave functions of these more localized states typically extend over at least tens of atoms in the amorphous network[39–41]. A change in their occupation must then lead to structural relaxation in the surrounding network[40,41], as governed by the bonding or antibonding nature of the state.

Remarkably, and difficult to capture in simulations, the observed attempt frequencies shown in Fig. 3b span 16 orders of magnitude from 100 Hz to $10^{18}$ Hz. Actually, the attempt frequency can be eight orders of magnitude higher when transitioning from state $i$ to state $j$ than for the opposite direction. To understand this, we move from the empirical Arrhenius equation to the transition state theory[42] and the Eyring equation[43] to express the transition rate in terms of the free energy difference $\Delta F = \Delta U - T\Delta S$ between the starting state at a landscape valley and the transient state at a landscape saddle point:

$$\nu = \nu_{00} \exp\left(\frac{\Delta S}{k_B}\right) \exp\left(-\frac{\Delta U}{k_B T}\right), \text{ with } \nu_{00} = \kappa \frac{k_B T}{h}, \quad (3)$$

where $h$ is Planck's constant, $\Delta S$ is the entropic contribution and $\Delta U$ is the (internal) energy contribution to the free energy. In applying equation (3), we set the transmission coefficient to $\kappa = 1$, that is, we neglect dynamical recrossing events.

## Towards a potential energy landscape picture

We repeat the fit of the temperature-dependent transition rates, now using the Eyring equation. Figure 4a plots the entropic and energetic contributions to the free energy barriers at the average of the temperature interval in which the corresponding transition rate was observed. The entropic contributions are mostly negative over the entire temperature range, effectively lowering an attempt frequency prefactor from $\nu_{00} \approx 5$ THz (at 200 K). For comparison, localized vibrational modes in amorphous GeTe are found around 4.8 THz (ref. 44). The lowering of

the attempt frequency is the reason why transitions with small energy barriers below 0.4 eV can dominate the noise dynamics on the timescales of our experiments.

To finally visualize the resulting landscape, information about its barriers alone is insufficient. In the potential energy landscape picture, the observed states correspond to basins of attraction of inherent structures (or rather 'metabasins' composed of multiple inherent structures that are separated by small energy barriers compared with $k_B T$ (ref. 45)). Entropy then quantifies the size of a (meta)basin and the state energy as a (weighted) average basin depth. Computer simulations suggest that not only the average energy but also the size of metabasins within the same system can vary widely. For example, Mocanu et al. find metabasins with between 6 and 600 inherent structures in their model glass system[16]. Here we use the fact that the (ergodic) Markov chains converge to a unique stationary state distribution $\pi$ within the experimental observation window (Fig. 4b) for a five-state system. The probability $\pi_i$ of observing the system with $n$ states in basin $i$ at any point in time is then dependent on the (relative) free energy $\delta F_i = \delta U_i - T\delta S_i$ of the state and the partition function $Z$:

$$\pi_i(T) = \frac{1}{Z} \exp\left(-\frac{\delta U_i - T\delta S_i}{k_B T}\right), \text{ where } Z = \sum_i^n \exp\left(-\frac{\delta U_i - T\delta S_i}{k_B T}\right). \quad (4)$$

From the temperature dependence of the state distribution function (Fig. 4c), we can, therefore, derive information about the average depth and width of a basin. For clarity, we illustrate a representative system with five states (Fig. 4d, bottom) as a fully interconnected five-state landscape, which can be clearly displayed in two spatial dimensions. Combining the information about barriers and basins reveals a self-consistent picture of a diverse landscape. The landscape is defined by the free energies of the five basin states (fit values are obtained from Fig. 4c) and the transition states in the middle between each pair of states, and is linearly interpolated in between. Energies correspond to landscape heights and entropies are linearly mapped to widths.

**a**

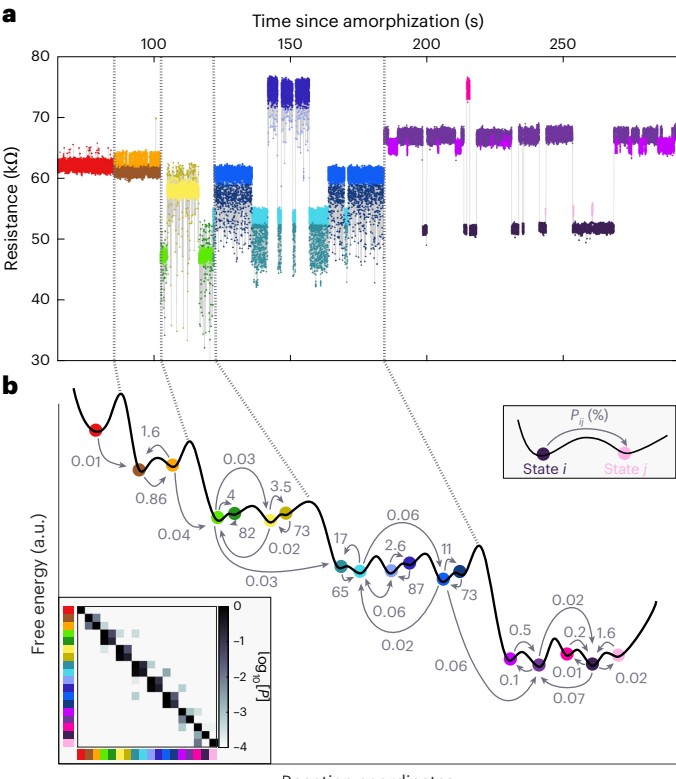

**Fig. 5 | Discrete nature of resistance drift. a**, Resistance time series of a PCM cell following amorphization with a lower-amplitude voltage pulse, resulting in a small amorphous volume without the subsequent application of external heating pulses. The resistance trace reveals rare, abrupt transitions between segments (separated by grey dashed lines in the figure), each characterized by distinct noise behaviour. As before, the data points are assigned to states and colour coded based on the most likely state sequence through the time series, as predicted from a Markov model fit. **b**, Effective free energy surface inferred from a hidden Markov model fit to the time trace shown in **a**. The landscape reflects the observed states and transition probabilities between them, with barrier heights approximately scaled to represent the latter. The inset in the bottom left shows the full transition probability matrix for the observed time window.

The red line traces the system's path through this landscape over a 2-s time period, as observed in the experiment (Supplementary Video 1 shows an animation). The dynamics and underlying landscape are characterized by high-energy states with large entropies and short sojourn times on one hand and low-energy states with small entropies and long sojourn times on the other hand. In transitioning between basins, the system must 'squeeze' through a more narrow part of the landscape as it overcomes some energy barriers of <0.2 eV. We note that such a narrowing in the landscape explored by the system at the transition state is predicted in the framework of multidimensional transition state theory[46]. When the temperature is increased, the system eventually tends to sample relatively larger barriers with even more narrow transition states compared with the (meta)basin size, though the general characteristics of the landscape remain the same.

## Evolving noise dynamics during physical ageing

Resistance drift and low-frequency noise in PCM devices are often attributed to related processes. If drift reflects physical ageing towards deeper regions of the energy landscape, then the metastable-state fluctuations underlying noise should also evolve in time. Our quasi-equilibrium analysis shows that entropic narrowing can strongly restrict interbasin transition rates, limiting configurational

exploration even for modest barriers[22]. This suggests that during ageing, the system transitions slowly between larger metabasins, accompanied by changes in the noise dynamics. To test this, we create a device state that exhibits physical ageing by using a low-amplitude amorphization pulse to produce a smaller glass volume without subsequent heater pulses. The resistance trace then shows rare transitions between segments, each characterized by more frequent fluctuations between resistance levels (Fig. 5a). These dynamics are again captured by a hidden Markov model (Fig. 5b), yielding transition probabilities between all the observed states. Although sparse transitions (and the lack of temperature-dependent data) preclude a full thermodynamic decomposition, the transition matrix still enables an informed free energy surface describing the evolving noise dynamics. In particular, the transitions are not strictly directed towards higher resistances. Drift, therefore, does not proceed monotonically, but via probabilistic moves across a rugged landscape shaped by entropic constraints.

To conclude, the noise dynamics of a nanoscopic volume of GeTe glass in a memristive device are characterized by fluctuations between few resistance states with a clear separation of fast and slow timescales, indicative of the underlying potential energy landscape. Modelling the experimental data with a hidden Markov model allows a reliable identification of states and transition rates. Although the resulting activation energies agree with energy barriers observed in computer simulations, they only provide an incomplete landscape picture, which alone cannot explain the measured dynamics. At the typical phonon (terahertz) attempt frequencies, the transition rate across these many small barriers (<0.4 eV) would be too fast to be observed in our experiments. However, the observed attempt frequencies span many orders of magnitude down to 100 Hz. Our results lead to a picture of a diverse potential energy landscape in which the rate-limiting narrowing of the energy landscape at a saddle point compared with the basin is quantified through a negative entropic contribution to the transition rate.

A practical limitation of the current approach is that the dynamics must produce discrete, temporally resolvable transitions in the measured observable, which requires a sufficiently small active volume. Where this holds, time-resolved resistance traces provide direct access to metastable states and barrier-crossing events. The direct validation of the reconstructed free energy landscape by atomistic simulations remains challenging because accessible timescales are limited and entropic effects are difficult to capture, even with metadynamics methods. Progress in machine-learned potentials, coarse-graining and high-performance computing should enable increasingly direct comparisons between experiments and simulations.

Although this study focuses on a prototypical PCM device, the methodology and insights should be broadly applicable to other memristive systems, for example, resistive random-access memory devices[47]. In many of these, resistance changes are governed by atomic-scale structural transitions under non-equilibrium conditions where stochastic fluctuations shape device dynamics. Our approach may provide a powerful framework for analysing such dynamics and quantifying the thermodynamic character of accessible states across diverse memory technologies. In these devices, noise is not necessarily detrimental, but can support stochastic learning rules[48], enhance exploration during optimization[49] and facilitate probabilistic inference in Bayesian in-memory computing systems[50].

## Online content

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

## Methods

### Device fabrication

For the fabrication of the the buried microheater, a 50-nm-thick platinum layer is evaporated onto 200-nm thermal silicon dioxide and patterned using optical lithography and a lift-off process. It is buried underneath 25 nm of evaporated tantalum that is subsequently oxidized in an oxygen atmosphere, forming 50 nm of tantalum oxide. An additional 200 nm of silicon dioxide is deposited and polished (chemical mechanical polishing) down to 100 nm.

For the GeTe cell, 30-nm GeTe is sputter deposited and capped with a thin layer of silicon dioxide to prevent oxidation during the following fabrication steps. The layer stack is patterned into the butterfly shape using HSQ resist, electron-beam lithography and ion milling. To add electrical connections, vias are etched into the layer stack, 40-nm tungsten is deposited and patterned again by electron-beam lithography and ion milling. A 5-kΩ resistance is added by winding the tungsten lead to protect the low-resistance crystalline cell against large current spikes. Finally, the entire structure is encapsulated in 80 nm of silicon dioxide. Gold contact pads are added using optical lithography for contacting with probe heads.

### Electrical measurements

The sample chip with hundreds of devices is placed in a liquid-nitrogen-cooled cryogenic probe station (Janis ST-500), which sets the base temperature during the noise measurements with millikelvin accuracy. An individual device is contacted with a Multi-$|Z|$ probe (FormFactor) mounted on piezo-stages. The microheater is connected to ground on one side, and on the other side, to a source measure unit (Keithley 2612B) and a pulse generator (Keysight B1525A) over a relay. The GeTe cell is connected to a signal generator (Agilent 81160A) on one side, and the transmitted current is amplified (FEMTO DLPCA-200 transimpedance amplifier) and recorded at a lock-in amplifier (Zurich Instruments UHFLI).

At the start of a measurement series, a centre fraction of the GeTe volume is amorphized by applying a 7-V, 50-ns rectangular (2-ns leading and trailing edges) pulse to the GeTe cell. To measure the resistance, a small 0.1-V, 10-kHz a.c. bias is applied and the transimpedance-amplified signal is measured using the lock-in amplifier with a 1-kHz bandwidth, which limits the time resolution in the time-resolved noise measurement. We choose 10 kHz as the carrier frequency, since it is above the $1/f$ noise contributions of the transimpedance and lock-in amplifiers and below the characteristic frequencies of other laboratory-specific external noise sources. The measurement of the 1-MΩ resistor (Fig. 1e) is close to the noise floor of the setup. The measured amorphous cell resistance is frequency independent from the d.c. limit up to gigahertz frequencies at which parasitic capacitances in parallel to the GeTe cell overshadow the signal.

The cell is then partially recrystallized to a value of around 1 MΩ. For this purpose, consecutive 450-K, 10-µs heating pulses are applied to the microheater. For the following time-domain noise measurement over 1,000 s, the real part of the lock-in signal is sampled at 20 kHz and streamed to a computer. Oversampling the signal with respect to the lock-in bandwidth by a factor 10 avoids aliasing effects. The recorded signal is decimated and low-pass filtered to 2 kHz, which further suppresses external higher-frequency noise contributions.

The time-domain resistance fluctuations are measured in this way at temperatures ranging from 100 K to 380 K. A specific temperature is set by applying a defined bias to the calibrated microheater with the source measure unit. In between the temperature steps, the microheater is switched off and the resistance is remeasured at the cryostat base temperature. If an increase in the time-averaged resistance is detected, which would be indicative of structural relaxation and an escape of the system from the before-investigated region of the landscape, the measurement is aborted and the cell is recrystallized entirely.

Adding a small d.c. offset of 10–50 nA (corresponding to a d.c. voltage bias comparable in amplitude to the a.c. bias) has little effect on the kinetics and noise power spectra. For higher biases, we observe an acceleration of the kinetics consistent with a temperature increase in the nanoscopic volume rather than a tilting of the energy landscape, as observed elsewhere[31]. Although we believe that the role of external fields for the shape of the energy landscape is very interesting, the application of external fields ideally would be decoupled from the measuring process. For instance, the additional electric field could be applied through a gate electrode to decouple temperature from field effects.

### Markov modelling

We fit hidden Markov models to the experimental data using the Baum–Welch algorithm, as implemented in the Python package pomegranate[35]. The state emissions are modelled as normal distributions. Each fit is repeated 100 times with randomized starting parameters to avoid local likelihood maxima. Data points are assigned to states based on the most probable state sequence through the time series, which is calculated using the Viterbi algorithm implemented in pomegranate. We check the quality of the model description and estimate the number of observed states by calculating the (ordinary) pseudoresiduals, as described elsewhere[34].

### Construction of landscape visualization

To generate Fig. 4d, we combine the temperature-dependent transition rates between states, obtained from the fitted hidden Markov models, and the temperature-dependent stationary state distribution $\pi(T)$. The transition rates are analysed using the Eyring equation to extract, for each directed transition $i \rightarrow j$, the internal energy contribution $\Delta U$ and the entropic contribution $\Delta S$ to the corresponding free energy barrier $\Delta F = \Delta U - T\Delta S$. Relative basin free energies $\delta F_i = \delta U_i - T\delta S_i$ are obtained by fitting the temperature dependence of $\pi_i(T)$ to equation (4).

For landscape visualization, we select a temperature interval and amorphization cycle in which the noise dynamics are governed by five states. A fully connected five-state system can be clearly visualized in two spatial dimensions, enabling the interpretable depiction of a sampled section of the landscape. The basin energies $\delta U_i$ determine the vertical positions of the minima, whereas the entropic contributions $\delta S_i$ are mapped linearly to the basin widths. Barrier heights are set by the corresponding $\Delta U$ values, and barrier widths are mapped from $\Delta S$ in the same way. Landscape segments between the basin minima and saddle points are linearly interpolated. The experimentally observed state sequence at a temperature of 105 K within a 2-s interval is then plotted as the red trajectory across the reconstructed landscape. Supplementary Video 1 shows an animation of the same trajectory. Fluctuations within a basin are meant to illustrate the progression of time.

## Data availability

The study generated about 1.5 TB of data, which are available from the corresponding authors upon request.

## Code availability

The code for the analysis of all datasets is available from the corresponding authors upon request. The Markov model analysis uses the open-source Python package pomegranate[35].

## Acknowledgements

M.S. acknowledges support from the German Research Foundation (DFG) through the collaborative research centres Nanoswitches (CRC 917) and Intelligent Matter (CRC 1459) as well as the European Research Council (ERC-Grant 640003).

## Author contributions

S.W. performed the experiments and analysed the data for Figs. 1–4. X.T.V. designed the microheater structures. J.B. performed the experiment and analysed the data for Fig. 5. S.W., J.B., N.H., N.V. and M.S. contributed to the interpretation and compilation of the results in a series of discussions in the course of this study. S.W. wrote the manuscript with input from all authors. M.S. provided resources and acquired funding.

## Funding

## Competing interests

The authors declare no competing interests.

## Additional information

**Correspondence and requests for materials** should be addressed to Sebastian Walfort or Martin Salinga.

