## [Peer Review File · Nature Materials]

A free energy landscape analysis of resistance fluctuations in a memristive device

Corresponding Author: Professor Martin Salinga

Version 0:

Reviewer comments:

Reviewer #1

(Remarks to the Author)

The paper reports on an interesting analysis of the resistance noise in the amorphous state of the phase change compound GeTe. In small amorphous samples, discrete Lorentzian features superimpose to the usual $1/f$ spectrum, as previously shown in similar materials (Ref. 63).

In the present paper, the authors develop a thorough analysis based on a Markov chain model that allowed extracting crucial information on the dynamics of the system, such as the attempt frequencies and the activation energies for transitions among different metabasins of the potential energy landscape.

To my knowledge, this is the first experimental measurements of the transition rates among metabasins in an amorphous material. Moreover, for the material chosen here, this information is also of practical relevance because the transitions responsible for the noise in resistance are believed to rule also the resistance drift which has an impact on the operation of memristors devices. For this reason, I think that the paper deserves publication on Nature Materials. The paper is clearly written, I list below a few minor issues to be addressed on resubmission.

- 1) On pag. 2 line 28. "... in part because the absence of long range atomic order .." I appreciate that the authors specify "in part", but this sentence suggest the higher resistivity of the amorphous phase with respect to the crystal is due to electron localization. Since conductance in phase change materials is typically described by the Poole-Frenkel model, it is certainly true that localization plays a role, but the main source of the contrast in resistivity is due to a shift of the Fermi level. The crystal is a degenerate semiconductor while the amorphous phase is a nearly intrinsic, albeit disordered, semiconductor.
- 2) Information on dashed pink lines in Fig 1e should be given in the figure caption
- 3) The $1/f^\alpha$ function with $\alpha=1.3$ does not fit very well the high frequency part of the spectrum in Fig. 1e. What happens for the usual value of $\alpha=1$?
- 4) On pag. 5 line 133, regarding the relation between Peierls distortion and gap opening, the authors refer to two articles (Refs. 22,64) that deal with different materials (Sb and GeSb). In a theoretical paper by Cobelli et al., Phys. Rev. Mater. 5, 045004 (2021), the same relation is discussed for the GeTe compound addressed in the present article.
- 5) On pag. 9 line 26 "... over tens or hundreds of atoms ..." refer to simulations with a cell containing just a few hundred atoms in which it is very difficult if not impossible to qualify as localized a state spread over "hundreds of atoms".
- 6) The analysis in Fig. 4 refers to a set of measurements interpreted with a Markov chains with 5 states. Perhaps some comments could be given on why a different set of measurements has been chosen for this figure with respect to those analyzed with 6 states in Fig. 2-3.
- 7) The Eyring formula (eq. 3) should in principle contains also a transmission coefficient which is here taken as one for all processes (no recrossing events). That's fine, but it should be stated explicitly as this is an additional approximation considering the complexity of the system.

Reviewer #2

(Remarks to the Author)

The authors analyze low frequency noise in a device consisting of a GeTe microscale resistor with a Pt joule heater used for resistance state programming. In particular, they use a Markov chain based analysis to analyze the fluctuations in resistance in high and low resistive states. By performing this analysis as a function of temperature, the authors are able to derive a

series of Arrhenius plots for a set of 6 resistance states consistent with their model. This analysis departs from the typical 2 state model with one potential barrier. The authors do not describe what these states are, but presumably these are various local atomic arrangements that define the local rate of charge transfer.

While I agree with the authors that understanding noise in analog memristive devices is important to implementing these emerging technologies in energy efficient computing, I disagree with their key claim that the results are widely applicable to other non volatile memory technology. I also find the lack of predictive analysis for PCM in terms of analog state density, retention and scaling to be an important deficit. For these reasons, I believe the paper is not currently suitable for publication in Nature Materials.

1. The authors claim that their approach is broadly applicable, without any evidence. For example, low frequency noise mechanism have been extensively studied in filament forming resistive memory (RRAM). In these devices, LFN can be well described by an electronically active defect in the dielectric surrounding the filament and the noise originates from the trapping and detrapping of carriers in the semiconducting medium surrounding the filament. For this reason, noise can be substantially reduced in ReRAM by increasing the electronic conductivity of the filament, thus enabling it to block the effects of the nearby traps (see K. Sugawara, Low-Frequency-Noise Spectroscopy of TaOx-based Resistive Switching Memory, *Adv. Electron. Mater.* 2022, 8, 2100758; and Rao et al., Thousands of conductance levels in memristors integrated on CMOS, *Nature* 615, 2023, p.823). This mechanism is substantially different from PCM, where transport, at least in the amorphous phase is possibly better described by Marcus theory, rather than band theory. Considering other emerging non volatile memories like STT MRAM, FeRAM and ECRAM, where charge transport and switching mechanisms are even further removed, it is hard to see how the approach described in this submission is widely applicable.

2. There is a lot of published data on LFN in PCM devices (see for example Giannopoulos et al., IEDM proceedings 2018). It could be interesting for the authors to use their analysis to make predictions regarding device performance in terms of noise and retention and how these will scale and compare that to published reports.

Reviewer #3

(Remarks to the Author)

In this work, the authors reported a free energy landscape analysis of resistance fluctuations in a phase-change memory device. The authors tried to provide a more quantitative picture of the high-dimensional energy landscape for fragile glass based on experimental measurements. This is a very challenging topic. In the amorphous materials community, the “energy landscape versus reaction coordinate” plot is frequently used, but these plots are simple hand drawings with no solid support of experimental data. Hence, this work provides very valuable insights on the atomistic understanding of glass relaxation and dynamics. The authors chose to work on a prototype phase-change material, GeTe, and designed a bridge-type butterfly-shape device for electrical measurements. Starting from a freshly as-deposited amorphous phase or a pulsing-induced melt-quenched amorphous phase, the resistance value of amorphous GeTe increases with time spontaneously, following a power law. Upon heating, GeTe crystallizes into a rhombohedral phase, which results in a drastic decrease in resistance. They measured the resistance values of the GeTe device after voltage pulse programming and/or under different holding temperatures and fitted the data against Markov models. They were able to provide transition rate / energy / entropy versus temperature, and provide a very interesting energy landscape plot with five basins in Fig. 4d. Hence, I recommend its publication at *Nat Mater*, if the authors can address the following points in a convincing way:

1>I am not sure if “memristive” is a suitable term. In particular, the authors claimed “It can provide new insights of all kinds of memristive materials”. Quite some memristive materials are based on conducting channels, spin flips, crystal-to-crystal transition or even organic materials, and I doubt if the method developed here can really help elucidate the switching process in these materials. In these materials, there is also no notable resistance drift issue.

2>I would suggest the authors to shorten the discussions on noise/variability related applications, which are not really true. Devices with non-ideal properties can enable some applications, but regarding commercialization of neuromorphic in-memory computing, high-precision computing is the real goal, because high-precision calculations cover the vast majority of computing tasks. This also holds for phase-change memory and phase-change in-memory computing, as resistance drift and nucleation-induced crystallization randomness are the bottlenecks for industrial development.

3>Instead, it would be more useful to discuss the limitation of the current approach in probing the energy landscape. Also, what else can one do to validate the sketched energy landscape? Will MD calculations help? In the field of amorphous materials, amorphous Si, SiO₂, GeS₂ and bulk metallic glasses are frequently studied, can one expect to use the current setup for the investigations of these amorphous materials?

4>The authors need to provide more details on how they obtained Fig. 4d as Supplementary Notes. A Supplementary Video highlighting the transition between different basins would also be very helpful.

5>Fig. 3a/3b contain many points, and are difficult to read. There is no label for each set of datapoints.

6>Do the “states” in Fig. 2a and Fig. 4b represent the same things? The color coding looks similar, but there are 6 states in Fig. 2a and 5 states in Fig. 4b. I also don't understand whether these states represent amorphous GeTe (of the same volume) in different energy basins or just different device states with a varied amorphous-to-crystal ratio of GeTe?

7>Are the GeTe thin films covered by any capping layer? Will evaporation and oxidation be a problem after multiple pulsing and long-term placement in air?

8>It would be useful to provide the width of the GeTe center area. Have the authors tried to vary this size? If not, some comments on the potential impact could be useful.

9>What is the unit of Fig. 1d? It is not clear to me how large is the variation. The original resistance values could be provided in the supporting information.

10>Too many references were cited in the main text, in total 88. Please remove the less relevant ones or move some of them

to the supporting information. For instance, quite some references of in the introduction (57 refs) are not necessary. Please reduce the number of the reference cited in the main text to 50 or below.

Version 1:

Reviewer comments:

Reviewer #1

(Remarks to the Author)

The authors have replied satisfactorily to all the issues raised by the reviewers and have revised the paper accordingly. The article can be published in the present form.

I just bring to the authors attention that units are missing in the vertical axis of Fig. 5b.

Reviewer #2

(Remarks to the Author)

I am satisfied with the authors' response and changes made to the manuscript and feel it is suitable for publication in Nature Materials.

Reviewer #3

(Remarks to the Author)

The authors have replied to my comments in a convincing way. They have also revised their manuscript thoroughly and included additional analyses and discussions. I think that the manuscript is now suitable for publication at Nat Mater.

Colour code:

- Reviewers' comments are in black.
- The authors' replies are in blue.
- Changes to the manuscript are underlined in yellow.

Reviewer 1

The paper reports on an interesting analysis of the resistance noise in the amorphous state of the phase change compound GeTe. In small amorphous samples, discrete Lorentzian features superimpose to the usual $1/f$ spectrum, as previously shown in similar materials (Ref. 63). In the present paper, the authors develop a thorough analysis based on a Markov chain model that allowed extracting crucial information on the dynamics of the system, such as the attempt frequencies and the activation energies for transitions among different metabasins of the potential energy landscape. To my knowledge, this is the first experimental measurements of the transition rates among metabasins in an amorphous material. Moreover, for the material chosen here, this information is also of practical relevance because the transitions responsible for the noise in resistance are believed to rule also the resistance drift which has an impact on the operation of memristors devices. For this reason, I think that the paper deserves publication on Nature Materials. The paper is clearly written, I list below a few minor issues to be addressed on resubmission.

We would like to thank Reviewer 1 for this positive and considered evaluation of the manuscript. We will address each minor issue in the following. Furthermore, an additional Figure (5) now makes the link between resistance noise and resistance drift more explicit (line 302).

1) On pag. 2 line 28. " ... in part because the absence of long range atomic order .." I appreciate that the authors specify "in part", but this sentence suggest the higher resistivity of the amorphous phase with respect to the crystal is due to electron localization. Since conductance in phase change materials is typically described by the Poole-Frenkel model, it is certainly true that localization plays a role, but the main source of the contrast in resistivity is due to a shift of the Fermi level. The crystal is a degenerate semiconductor while the amorphous phase is a nearly intrinsic, albeit disordered, semiconductor.

We agree with Reviewer 1 that our statement does not provide a complete picture of the origin of the contrast in resistivity between crystalline and amorphous PCMs. For the benefit of brevity in the Introduction, we have removed this part of the sentence (line 26). This is also more consistent with our agnostic stance on the origin of any resistance change in the latter part of the manuscript.

2) Information on dashed pink lines in Fig 1e should be given in the figure caption

We have added a description of the two Lorentzians with time constants τ_i in both the caption and legend of Figure 1e.

3) The $1/f^\alpha$ function with $\alpha=1.3$ does not fit very well the high frequency part of the spectrum in Fig. 1e. What happens for the usual value of $\alpha=1$?

We thank Reviewer 1 for this remark. In Fig. 1e, the $1/f^{1.3}$ line is intended purely as a guide to the eye, illustrating that the large-volume glass exhibits a broad, fairly featureless $1/f$ -like background

in contrast to the discrete Lorentzians seen for the small glass volume. Typical noise exponents of amorphous semiconductors including phase-change materials span a range of $\alpha \approx 0.8\text{--}1.4$ (e.g. Refs. [21, 28, 29] and reference in the next paragraph), and our spectrum is consistent with this variability. Using $\alpha \approx 1$ instead of $\alpha \approx 1.3$ does not change the interpretation and describes the data well within the uncertainty at high frequencies. To avoid any misunderstanding, we have added a brief statement clarifying this point in the figure caption.

We also note that even under nominally identical programming conditions, cycle-to-cycle variability in PCM devices leads to a large spread of power spectral density exponents. A recent study by IBM (Lombardo, Davide GF, et al. "Cryogenic In-Memory Computing with Phase-Change Memory." arXiv preprint arXiv:2509.22511 (2025), Fig. 6) reports values ranging from $\alpha \approx 0.9$ to $\alpha \approx 1.5$ across repeated reset cycles. This reinforces the idea that individual fitted exponents should not be over-interpreted. They reflect the stochastic variation of the device states rather than fixed material constants. In line with this, we deliberately avoid attributing physical meaning to individual α -values, just as in our time-domain analysis, where we do not overemphasise the significance of individual activation energies or attempt frequencies.

4) On pag. 5 line 133, regarding the relation between Peierls distortion and gap opening, the authors refer to two articles (Refs. 22,64) that deal with different materials (Sb and GeSb). In a theoretical paper by Cobelli et al., Phys. Rev. Mater. 5, 045004 (2021), the same relation is discussed for the GeTe compound addressed in the present article.

We thank Reviewer 1 for pointing this out. We agree that the work of Cobelli et al. (Phys. Rev. Mater. 5, 045004 (2021)) is directly relevant, as it analyses the connection between Peierls distortion and band-gap opening specifically in GeTe. We now explicitly cite this reference (line 132).

5) On pag. 9 line 26 ". . . over tens or hundreds of atoms ..." refer to simulations with a cell containing just a few hundred atoms in which it is very difficult if not impossible to qualify as localized a state spread over "hundreds of atoms".

We thank Reviewer 1 for this observation. Our statement was intended to convey that even the more localized electronic states in amorphous GeTe are not confined to a single atom but typically involve a cluster of multiple neighbouring atoms. To avoid any unintended implication of numerical precision, we have rephrased the sentence to: "The electronic wave function of these more localised states in PCMs is known to typically extend over at least several tens of atoms within the amorphous network." (line 241)

6) The analysis in Fig. 4 refers to a set of measurements interpreted with a Markov chains with 5 states. Perhaps some comments could be given on why a different set of measurements has been chosen for this figure with respect to those analyzed with 6 states in Fig. 2-3.

We thank Reviewer 1 for raising this point. The number of states obtained from the hidden-Markov analysis is not fixed, but it varies from one amorphisation (reset) cycle to the next and also depends on temperature for the same cycle. Figure 2a shows one representative example in which six states are resolved, whereas in other cycles or temperature ranges the observed resistance fluctuations can be described well with four or five states.

For the visualisation in Fig. 4b–d, we deliberately selected a representative 5-state system because a fully interconnected 5-state landscape can be visualised in two spatial dimensions. This choice is

purely for clarity of presentation and does not reflect any conceptual difference compared to the six-state example discussed in Fig. 2a. We have added a brief clarification to the manuscript: “For clarity, we illustrate a representative system with five states, as a fully interconnected five-state landscape can be displayed clearly in two spatial dimensions.” (line 281)

7) The Eyring formula (eq. 3) should in principle contains also a transmission coefficient which is here taken as one for all processes (no recrossing events). That’s fine, but it should be stated explicitly as this is an additional approximation considering the complexity of the system.

We thank Reviewer 1 for this helpful remark. In Eq. 3 we indeed have implicitly set the transmission coefficient $\kappa = 1$, i.e. we neglect dynamical recrossing events. The transmission coefficient is a dynamical correction factor accounting for very fast back-and-forth motion in the vicinity of a saddle. Such recrossings cannot be resolved within the temporal resolution of our experiment. We now state this explicitly in the manuscript (line 255).

Reviewer 2

The authors analyze low frequency noise in a device consisting of a GeTe microscale resistor with a Pt joule heater used for resistance state programming. In particular, they use a Markov chain based analysis to analyze the fluctuations in resistance in high and low resistive states. By performing this analysis as a function of temperature, the authors are able to derive a series of Arrhenius plots for a set of 6 resistance states consistent with their model. This analysis departs from the typical 2 state model with one potential barrier. The authors do not describe what these states are, but presumably these are various local atomic arrangements that define the local rate of charge transfer.

We appreciate the summary and remarks provided by Reviewer 2, as they show us that a few clarifications are necessary. The lateral dimension of the GeTe volume in our devices is on the order of 100 nm times 100 nm. Furthermore, our analysis does not use a fixed number of resistance states throughout the manuscript. The six-state model mentioned reflects one representative instance. In other measurement series and temperature ranges, the number of states can be different. These states represent distinct, statistically inferred resistance levels with characteristic transition rates within the active volume of the device. While we do not assign specific microscopic configurations to these states, they are consistent with different metastable atomic arrangements influencing the electronic transport. Our approach is thus not based on a material-specific model and is designed to capture experimentally observed resistance fluctuations in a minimally biased manner, agnostic to the atomistic details.

While I agree with the authors that understanding noise in analog memristive devices is important to implementing these emerging technologies in energy efficient computing, I disagree with their key claim that the results are widely applicable to other non volatile memory technology. I also find the lack of predictive analysis for PCM in terms of analog state density, retention and scaling to be an important deficit. For these reasons, I believe the paper is not currently suitable for publication in Nature Materials.

1. The authors claim that their approach is broadly applicable, without any evidence. For example, low frequency noise mechanism have been extensively studied in filament forming resistive memory (RRAM). In these devices, LFN can be well described by an electronically active defect in the dielectric surrounding the filament and the noise originates from the trapping and detrapping of carriers in the semiconducting medium surrounding the filament. For this reason, noise can be substantially reduces in ReRAM by increasing the electronic conductivity of the filament, thus enabling it to block the effects of the nearby traps (see K. Sugawara, Low-Frequency-Noise Spectroscopy of TaOx-based Resistive Switching Memory, Adv. Electron. Mater. 2022, 8, 2100758; and Rao et al., Thousands of conductance levels in memristors integrated on CMOS, Nature 615, 2023, p.823). This mechanism is substantially different from PCM, where transport, at least in the amorphous phase is possibly better described by Marcus theory, rather than band theory. Considering other emerging non volatile memories like STT MRAM, FeRAM and ECRAM, where charge transport and switching mechanisms are even further removed, its hard to see how the approach described in this submission is widely applicable.

We thank Reviewer 2 for raising the issue of generality and predictive value (the latter is addressed as a response to Remark 2). We would like to clarify that the energy landscape perspective we adopt is not specific to PCMs but applies to any system that exhibits fluctuations in some observ-

able. Here, different macroscopic observables necessarily reflect different microscopic configurations, whether structural, electronic, or both. This is a general physical description, not a material-specific assumption.

Our method requires only a time-resolved observable that is sensitive to such microscopic changes, in this case resistance, and does not rely on a predefined mechanism. As such, it can in principle be applied to a broad range of memory systems, provided time-resolved noise data are accessible. Memristive materials are ideal, as they are usually characterised by large changes in resistivity as a response to small changes in atomic/electronic structure. We acknowledge that different systems will exhibit different microscopic origins of noise, but these distinctions affect the shape and depth of the landscape, not the applicability of the framework itself. We believe our results demonstrate the usefulness of fitting a statistical model that takes both the resistance level and importantly the time-series nature of the observation into account, enabling us to reliably identify discrete states and track their evolution across temperature to quantify landscape features.

As pointed out by Reviewer 2, we agree that despite differences in material systems, the kind of RTN-like resistance fluctuations analysed in our study are not unique to PCMs and have also been observed and analysed in other memory technologies, including ReRAM. Visual inspection of the data presented by the two articles highlighted by Reviewer 2 suggests to us that also here resistance fluctuations can be described by a Markov model. However, the microscopic origin of the conductance noise for instance reported in Rao et al. (Nature 615, 823–829, 2023) is far from settled. This is already evident from the corresponding peer review file. One reviewer questioned whether the data truly support a conventional charge trapping interpretation and pointed to the possibility of structural fluctuations or filament rearrangements. In response, the authors clarified that they do not claim a definitive microscopic origin and acknowledged that the switching events likely involve structural relaxation processes. This exchange highlights that the physical nature of the observed fluctuations remains unresolved among experts of the field, and that multiple mechanisms, both electronic and structural in nature, may be involved. In the Supplementary Information, the authors go on to suggest that the electronic occupation of antibonding states leads to a relaxation of the surrounding atomic configuration, which is a mechanism that to us implicitly invokes an energy landscape of multiple metastable states. Our approach now offers a route to experimentally access the thermodynamic character of such states. By fitting a hidden Markov model to time-resolved resistance data across temperatures, one can extract not only the resistance levels and transition rates, but also the energetic and entropic contributions to the states themselves as well as the free energy barriers that govern both state occupation and dynamics. For one, this provides a complementary perspective to atomistic (usually frozen) structural models and offers more general insight into the physical nature of conductance states beyond the characteristics of a particular device, compared to making ad hoc assumptions about defect state distributions and attempt frequencies. We now explicitly cite the work by Rao et al. in the conclusion of the manuscript and emphasise that our approach may offer a systematic way to extract thermodynamic parameters from time-resolved conductance fluctuations (line 358).

2. There is a lot of published data on LFN in PCM devices (see for example Giannopoulos et al., IEDM proceedings 2018). It could be interesting for the authors to use their analysis to make predictions regarding device performance in terms of noise and retention and how these will scale and compare that to published reports.

We thank Reviewer 2 for pointing us to this study. We agree that low-frequency noise (LFN) in PCMs has been studied and cite several relevant prior works in our manuscript. The Giannopoulos et al. paper uses the projected memory concept developed by IBM, which introduces an additional parallel resistor to reduce noise/drift contributions from the amorphous region. While this strategy does lower the measured resistance noise power spectrum, it comes at the cost of increased device complexity and, more importantly, reduced resistance contrast, which is a well-understood trade-off. In their approach, the device architecture is thus modified to suppress conductance variability. In contrast, our approach aims to understand the thermodynamic origins and statistical properties of these fluctuations from a fundamental perspective.

Therefore, we believe the predictive power of our approach is best demonstrated through the addition of Fig. 5 and the accompanying paragraph to the manuscript (line 302), where we intentionally employ low amplitude voltage pulses to create a smaller amorphous volume directly during the reset process. This adjustment eliminates the need for subsequent additional external heating pulses and places the system in a regime where both fluctuations between discrete resistance levels are observable and physical ageing occurs. In this regime, the noise characteristics evolve discontinuously over time as the system undergoes physical ageing. Based on barrier heights from simulation studies (Ref. 21) one should expect that interbasin transitions in GeTe occur on timescales too fast to be observable in our experiments. Yet, we are able to directly capture such transitions on a seconds timescale, as predicted by the proposed entropic narrowing at the transition state. The physical ageing process is accompanied by drastic changes in the noise characteristics, as the system transitions from one metabasin to another. Importantly, these dynamics can again be described using a hidden Markov model, enabling the extraction of transition probabilities between all observed states. While these single-shot transitions naturally carry uncertainty and preclude a full thermodynamic decomposition, we can still construct an informed effective free energy surface based on the measured transition probability matrix. In this way, we can make experimentally grounded predictions about physical ageing dynamics as the systems evolves through the energy landscape, which remains difficult to access in atomistic simulations given current computational limitations.

Finally, we would like to emphasise that our work goes far beyond the conventional interpretation of low-frequency noise as a superposition of simple Lorentzian fluctuators or RTN arising from a single, dominant two-state defect. Instead, our method extracts the full temperature-dependent transition network between all experimentally accessible metastable states, providing state-resolved energies and entropies. These are quantities that are fundamentally inaccessible to standard PSD-based approaches. This analysis reveals that the large spread of apparent attempt frequencies arises naturally from pronounced entropic narrowing at the transition states. In this way, we progress from a phenomenological description of noise to the experimental mapping of the underlying thermodynamic structure. While the microscopic origins of fluctuations vary across memory technologies, any system in which resistance is sensitive to atomic-scale rearrangements can, in principle, be analysed using the same framework. Our method therefore offers a general approach to extracting thermodynamic descriptors directly from time-resolved noise data.

Reviewer 3

In this work, the authors reported a free energy landscape analysis of resistance fluctuations in a phase-change memory device. The authors tried to provide a more quantitative picture of the high-dimensional energy landscape for fragile glass based on experimental measurements. This is a very challenging topic. In the amorphous materials community, the “energy landscape versus reaction coordinate” plot is frequently used, but these plots are simple hand drawings with no solid support of experimental data. Hence, this work provides very valuable insights on the atomistic understanding of glass relaxation and dynamics. The authors chose to work on a prototype phase-change material, GeTe, and designed a bridge-type butterfly-shape device for electrical measurements. Starting from a freshly as-deposited amorphous phase or a pulsing-induced melt-quenched amorphous phase, the resistance value of amorphous GeTe increases with time spontaneously, following a power law. Upon heating, GeTe crystallizes into a rhombohedral phase, which results in a drastic decrease in resistance. They measured the resistance values of the GeTe device after voltage pulse programming and/or under different holding temperatures and fitted the data against Markov models. They were able to provide transition rate / energy / entropy versus temperature, and provide a very interesting energy landscape plot with five basins in Fig. 4d. Hence, I recommend its publication at Nat Mater, if the authors can address the following points in a convincing way:

We thank Reviewer 3 for the thoughtful summary and the positive assessment of our work. We appreciate the recognition of the challenges involved in mapping the free energy landscape of disordered materials and are encouraged that our approach is seen as a valuable contribution to the experimental investigation of glass dynamics.

1) I am not sure if “memristive” is a suitable term. In particular, the authors claimed “It can provide new insights of all kinds of memristive materials”. Quite some memristive materials are based on conducting channels, spin flips, crystal-to-crystal transition or even organic materials, and I doubt if the method developed here can really help elucidate the switching process in these materials. In these materials, there is also no notable resistance drift issue.

We thank Reviewer 3 for raising this concern. We agree that the term memristive materials can encompass a wide range of systems, including those based on filament formation, spin flips, or other structural phase transitions. Our intention was not to suggest that all memristive systems share the same microscopic switching mechanism, but rather to point out that these systems exhibit a common feature, i.e. small changes in the atomic, electronic, or vibronic structure lead to measurable changes in device resistance. (In the Abstract we exchanged “resistance swichting” with the more fitting “resistance change” and made further small adjustments in the two final sentences) From an experimental materials science perspective, this makes resistance, and its fluctuations over time, a very attractive experimental observable. It allows us to probe microscopic dynamics using a straightforward electrical measurement that locally senses the active device volume. Our analysis then rests on the general framework of a free energy landscape, which assumes only that different macroscopic resistance values correspond to different microscopic configurations. We believe this is a powerful perspective as it avoids ad hoc assumptions about combinations of defect distributions and attempt frequencies, which are often used to fit specific datasets but do not generalise well across materials, operating conditions, or even switching cycles.

Instead, we propose to fit a statistical model that captures both the resistance levels and the temporal

structure of their transitions. Through their temperature dependence we can disentangle competing energetic and entropic contributions to states and barriers, potentially offering much more general insights into the dynamics and stability of memristive states. We believe this type of analysis to be particularly interesting for nanoscale devices, where a small active volume is coupled to a large heat bath and driven by electrical inputs. In such settings, it may help disentangle intrinsic material dynamics from those induced by the environment. Understanding the origin of these dynamics could ultimately enable more targeted engineering of energetic and entropic contributions.

Through the addition of Figure 5 and accompanying paragraph in the manuscript (line 302), we demonstrate how the Markov model description is able to capture the evolving noise dynamics during resistance drift in PCM devices. In this case, drift reflects a gradual relaxation of the glass toward the supercooled liquid and corresponds to a slight slope in the energy landscape. However, this does not limit the generality of our approach. In systems without such directed drift, the dynamics can even be easier to analyse. We hope these additional explanations helped to clarify in what sense we think that our framework may offer a useful perspective to other memristive materials communities.

2) I would suggest the authors to shorten the discussions on noise/variability related applications, which are not really true. Devices with non-ideal properties can enable some applications, but regarding commercialization of neuromorphic in-memory computing, high-precision computing is the real goal, because high-precision calculations cover the vast majority of computing tasks. This also holds for phase-change memory and phase-change in-memory computing, as resistance drift and nucleation-induced crystallization randomness are the bottlenecks for industrial development.

In line with the suggestion, we limit the discussion of noise-enabled applications to a single sentence (line 364) in the conclusion section and clearly specify the context in which stochasticity may offer functional advantages, citing currently explored directions such as Bayesian in-memory computing (e.g., Dalgaty, T. et al., Nature Materials, 2025). Additionally, we have amended the final sentence of the abstract to make it clearer that the potential benefits of noise are context-specific and relevant to certain computing paradigms.

3) Instead, it would be more useful to discuss the limitation of the current approach in probing the energy landscape. Also, what else can one do to validate the sketched energy landscape? Will MD calculations help? In the field of amorphous materials, amorphous Si, SiO₂, GeS₂ and bulk metallic glasses are frequently studied, can one expect to use the current setup for the investigations of these amorphous materials?

We fully agree that a discussion of the limitations of our present approach and of possible avenues for external validation strengthens the manuscript, and we have now added a brief discussion in the conclusion (line 341). Our analysis relies on the fact that the resistance dynamics can be expressed as discrete, well-separated transitions between a number of metastable states. This requires that (i) the active resistive volume is sufficiently small so that internal rearrangements can be temporally resolved as individual events, and (ii) the observable (resistance) is sufficiently sensitive to these rearrangements. In larger volumes, many fluctuators are active simultaneously and the individual transitions merge into a featureless $1/f$ background, precluding the type of state-resolved analysis pursued here. Moreover, the hidden-Markov-model does not give spatial information about the internal rearrangements, but infers about the thermodynamics of the underlying energy landscape.

Molecular dynamics (MD) simulations would indeed be an ideal means of validating the experimen-

tally inferred landscape. However, even with machine-learned interatomic potentials, MD remains limited in the accessible timescales, which is several orders of magnitude shorter than the timescales that govern the experimentally observed transitions. Enhanced-sampling approaches such as metadynamics, transition-path sampling, or nudged elastic band calculations can complement MD by resolving reaction pathways and estimating barrier heights. However, they struggle to capture entropic contributions and often require predefined collective variables, which are not known a priori in disordered systems. We anticipate that continued progress in machine-learned potentials, coarse-graining strategies, and high-performance computing will narrow this gap, and enable increasingly direct comparisons between experiments and atomistic simulations. We have added a brief discussion to the manuscript (line 346).

We agree with the reviewer that many amorphous materials, such as amorphous Si, SiO₂, GeS₂, or bulk metallic glasses, are of great interest from the glass-physics and application perspective. However, these systems are not naturally suited to our method if their electrical resistivity is only weakly sensitive to small atomic rearrangements. In SiO₂ the relevant electronic states lie far from the Fermi level, so structural fluctuations do not produce measurable resistance changes. Metallic glasses face the opposite issue. The high density of electronic states at the Fermi energy leads to weak relative resistivity changes upon structural rearrangements. In both cases, electrical probing will likely struggle to resolve individual transitions. Probing their structural dynamics requires more involved, typically invasive techniques such as fluctuating speckle analysis in transmission electron microscopy, which offer high spatial- but has limited time resolution and do not naturally provide an observable with discrete state dynamics. One can maybe identify other suitable experimental observables, but the corresponding experimental technique needs to combine high spatial sensitivity with reasonable time resolution to observe smallest structural changes. In contrast, for all the above reasons memristive materials should be ideally suited because their electronic transport is extremely sensitive to minimal internal rearrangements, allowing thermally driven fluctuations on an atomistic level to be transduced into large, discrete resistance changes in a nanoscopic volume.

4) The authors need to provide more details on how they obtained Fig. 4d as Supplementary Notes. A Supplementary Video highlighting the transition between different basins would also be very helpful.

We thank the reviewer for the suggestion. In the revised version we have expanded the Methods section with a concise description of the procedure used to construct Fig. 4d (line 437). This addition brings together the key steps, that were described in the main text on pp. 9–10 and in the analysis of the temperature-dependent state distributions. Specifically, we explain (i) why a temperature interval where noise was governed by five states is chosen: this is the largest fully connected set of states that can still be clearly visualised in a two-dimensional landscape representation, (ii) how activation energies and entropies are obtained from the temperature-dependent transition rates through an Eyring analysis, (iii) how relative basin free energies are extracted from the temperature dependence of the stationary state distribution, and (iv) how these quantities are mapped to the landscape representation, where energies define the vertical axis and entropies are linearly mapped to basin and barrier widths. As recommended, we will also provide a Supplementary Video showing the system's experimentally observed trajectory through the reconstructed landscape.

5) Fig. 3a/3b contain many points, and are difficult to read. There is no label for each set of datapoints.

We thank the reviewer for this comment. Figure 3a shows a progression of Figure 2c, in which the individual transition rates between all the different states identified at each temperature are now shown. We make this link now more explicit in the caption of Fig. 3a. This naturally results in many data points, because each temperature window contains several metastable states and therefore multiple distinct state-to-state transitions. The figure caption and legend makes this explicit by stating how the colour of the marker face (state i) and the colour of the marker edge (state j) encode each directed transition.

In Fig. 3b, the large number of points reflects the full set of extracted (E_{ij}, ν_{ij}) pairs across all temperature intervals. The colours are not intended to label individual transitions; as stated in the caption, they simply distinguish different measurement series (separate crystallisation and re-amorphisation cycles). Their purpose is to illustrate the breadth of observed activation energies and, in particular, the wide distribution of attempt frequencies obtained across independent measurements, without placing too much visual emphasis on any specific transition pair. For readers seeking detailed information on individual transitions, including uncertainties, Supplementary Fig. S2 shows all (E_{ij}, ν_{ij}) combinations explicitly. We have now included this cross-reference also in the caption of Fig. 3b.

6) Do the “states” in Fig. 2a and Fig. 4b represent the same things? The color coding looks similar, but there are 6 states in Fig. 2a and 5 states in Fig. 4b. I also don't understand whether these states represent amorphous GeTe (of the same volume) in different energy basins or just different device states with a varied amorphous-to-crystal ratio of GeTe?

The Reviewer's first interpretation of a “state” accurately captures our usage of the term: A state corresponds to a basin in the energy landscape, characterised at a given temperature by a normally distributed resistance level and by its transition rates to the other states. These states are identified by fitting a hidden Markov model to the resistance–time traces. The same states can be observed across several temperature (Figs. 2c and 3a), which allows us to extract activation energies, attempt frequencies, and later the energetic and entropic contributions shown in Fig. 4a. However, if the temperature is increased further, the system eventually samples different regions of the landscape, as is clearly visible in Fig. 2c. When the device is recrystallised and re-amorphised, the amorphous structure changes and the accessible metastable configurations differ. Consequently, the number of resolvable states is not fixed, although, as we write in the manuscript, in practice 4–6 states describe the dynamics well within our experimental time resolution. For visual consistency we keep the colour coding uniform throughout the paper: at any given temperature and for a given amorphous volume, the states are ordered by increasing resistance and mapped onto the viridis colormap (blue to yellow). As we write in our response to Remark 4 and now expand on in the Methods section, we choose to show the landscape of a five-state system simply for visual clarity. Fig. 4b and 4c show how one can derive the necessary thermodynamic quantities about the basins themselves for the visualisation.

7) Are the GeTe thin films covered by any capping layer? Will evaporation and oxidation be a problem after multiple pulsing and long-term placement in air?

We share the Reviewer's concern regarding potential evaporation and oxidation of GeTe. To prevent these effects in our experiments, the GeTe films are capped immediately after sputter deposition without breaking vacuum, as described in the Methods section. Specifically, a thin SiO₂ capping layer is deposited in situ on top of the 30 nm GeTe film to avoid exposure of the surface to air during subsequent processing steps. After patterning and contact formation, the entire device structure is

further fully encapsulated in an additional SiO₂ capping (80 nm), ensuring long-term environmental stability. We therefore expect neither evaporation nor oxidation to influence the device characteristics measured under vacuum inside a cryostat. Our GeTe devices can be cycled between amorphous and crystalline several 10,000 times before any degradation of the contrast between set and reset states is observed. This is evidence that our precautions have been effective.

8) It would be useful to provide the width of the GeTe center area. Have the authors tried to vary this size? If not, some comments on the potential impact could be useful.

The width of the GeTe center region is varied from device to device ranging from approximately 50 nm to 100 nm. We have not carried out a systematic study in which the lateral width was intentionally varied as an independent parameter. However, we did explore the influence of the size of the amorphous active volume by varying the target resistance during stepwise re-crystallisation using the integrated Joule heater. Across these measurements, we did not observe any clear correlation between the device resistance (and thus amorphous volume) on the one hand and the extracted activation energies or attempt frequencies on the other. We now point to the corresponding Supplementary Fig. S3 in the caption of Fig. 3b.

9) What is the unit of Fig. 1d? It is not clear to me how large is the variation. The original resistance values could be provided in the supporting information.

We thank the Reviewer for pointing this out. In the revised version, we have added explicit tick labels on the y-axis of Fig. 1d as well as the average resistance values in the figure legend to make the magnitude of the fluctuations clear.

10) Too many references were cited in the main text, in total 88. Please remove the less relevant ones or move some of them to the supporting information. For instance, quite some references of in the introduction (57 refs) are not necessary. Please reduce the number of the reference cited in the main text to 50 or below.

We have removed references in the Introductions section that served to illustrate the breadth of the respective fields. The total number of references is now 50.

Colour code:

- Reviewers' comments are in black.
- The authors' replies are in blue.
- Changes to the manuscript are underlined in yellow.

Reviewer 1

The authors have replied satisfactorily to all the issues raised by the reviewers and have revised the paper accordingly. The article can be published in the present form. I just bring to the authors attention that units are missing in the vertical axis of Fig. 5b.

Reviewer 2

I am satisfied with the authors' response and changes made to the manuscript and feel it is suitable for publication in Nature Materials.

Reviewer 3

The authors have replied to my comments in a convincing way. They have also revised their manuscript thoroughly and included additional analyses and discussions. I think that the manuscript is now suitable for publication at Nat Mater.

We would like to thank all three reviewers for their careful evaluation of the manuscript and for their constructive feedback, which has helped improve the clarity and presentation of the work. We are grateful for the positive assessments. In response to Reviewer 1's final comment, we have updated Fig. 5b by explicitly labelling the vertical axis in arbitrary units. This is consistent with the Figure caption's description of the landscape as an effective representation scaled from transition probabilities.